# Osteopontin, Macrophage Migration Inhibitory Factor and Anti-Interleukin-8 Autoantibodies Complement CA125 for Detection of Early Stage Ovarian Cancer

**DOI:** 10.3390/cancers11050596

**Published:** 2019-04-28

**Authors:** Jing Guo, Wei-Lei Yang, Daewoo Pak, Joseph Celestino, Karen H. Lu, Jing Ning, Anna E. Lokshin, Zhongping Cheng, Zhen Lu, Robert C. Bast

**Affiliations:** 1Department of Obstetrics and Gynecology, Shanghai Tenth People’s Hospital, Tongji University School of Medicine, Shanghai 200072, China; camelguo2012@gmail.com (J.G.); mdcheng18@tongji.edu.cn (Z.C.); 2Department of Experimental Therapeutics, University of Texas M.D. Anderson Cancer Center, Houston, TX 77030, USA; newworldyang@gmail.com; 3Department of Biostatistics, University of Texas M.D. Anderson Cancer Center, Houston, TX 77030, USA; DPak@mdanderson.org (D.P.); jning@mdanderson.org (J.N.); 4Department of Gynecologic Oncology and Reproductive Medicine, University of Texas M.D. Anderson Cancer Center, Houston, TX 77030, USA; jcelesti@mdanderson.org (J.C.); khlu@mdanderson.org (K.H.L.); 5Department of Epidemiology, Pathology, Medicine, and Obstetrics/Gynecology and Reproductive Sciences, University of Pittsburgh, Pittsburgh, PA 15260, USA; loksax@UPMC.EDU

**Keywords:** ovarian cancer, osteopontin (OPN), macrophage migration inhibitory factor (MIF), CA125, IL-8 autoantibodies (IL-8 AAb), early detection

## Abstract

Early detection of ovarian cancer promises to reduce mortality. While serum CA125 can detect more than 60% of patients with early stage (I–II) disease, greater sensitivity might be observed with a panel of biomarkers. Ten protein antigens and 12 autoantibody biomarkers were measured in sera from 76 patients with early stage (I–II), 44 patients with late stage (III–IV) ovarian cancer and 200 healthy participants in the normal risk ovarian cancer screening study. A four-biomarker panel (CA125, osteopontin (OPN), macrophage inhibitory factor (MIF), and anti-IL-8 autoantibodies) detected 82% of early stage cancers compared to 65% with CA125 alone. In early stage subjects the area under the receiver operating characteristic curve (AUC) for the panel (0.985) was significantly greater (*p* < 0.001) than the AUC for CA125 alone (0.885). Assaying an independent validation set of sera from 71 early stage ovarian cancer patients, 45 late stage patients and 131 healthy women, AUC in early stage disease was improved from 0.947 with CA125 alone to 0.974 with the four-biomarker panel (*p* = 0.015). Consequently, OPN, MIF and IL-8 autoantibodies can be used in combination with CA125 to distinguish ovarian cancer patients from healthy controls with high sensitivity. Osteopontin appears to be a robust biomarker that deserves further evaluation in combination with CA125.

## 1. Introduction

Ovarian cancer is diagnosed at late stage (III, IV) in more than 70% of cases, contributing to poor patient outcomes. Over 22,530 women in the US will be diagnosed with ovarian cancer this year and about 13,980 women will die from the disease [1]. While five-year survival has improved over the last three decades, long term survival has changed little over the same interval and less than 30% for patients with late stage disease can be cured [2]. By contrast, when ovarian cancer is diagnosed in stage I, up to 90% of patients can be cured with conventional surgery and chemotherapy. Even in Stage II, up to 70% of patients survive five years with currently available therapy [2]. Consequently, survival might be significantly improved by detecting early stage ovarian cancer in a larger fraction of patients.

Attempts to detect ovarian cancer have utilized serum biomarkers, notably CA125, and imaging with transvaginal sonography (TVS). Used individually, neither CA125 nor TVS is sufficiently sensitive or specific for cost-effective screening [3]. Two-stage strategies have proven more promising, where rising CA125, judged by the risk of ovarian cancer algorithm (ROCA), has triggered TVS in a small fraction of participants, prompting surgery when imaging suggests possible cancer. Two clinical screening trials—the normal risk ovarian cancer screening study (NROSS) [4] and the United Kingdom Collaborative Study of Ovarian Cancer Screening (UKCTOCS) [5]—have achieved adequate specificity with three to four operations for each case of ovarian cancer detected. Both trials have detected a larger fraction of early stage (I–II) cases (40–64%) than observed at the time of conventional diagnosis (20–25%). With 200,000 participants, the UKCTOCS was powered to detect a survival advantage and in one pre-specified subset of women screened for seven years a 20% reduction in mortality was observed, although additional follow-up will be required to assure the accuracy of this estimate.

Whatever the outcome of the UKCTOCS, there is room for improvement in both phases of the two-stage strategy. CA125 is expressed by only 80% of ovarian cancers and is elevated in sera from approximately 65% of patients with early stage disease. Additional biomarkers will be required to detect a larger fraction of cases. Several serum biomarkers have shown promise for early detection in blood samples from ovarian cancer patients at the time of clinical diagnosis [4,5,6,7,8]. Both HE4 and CA72.4 can detect up to 16% of cases missed by CA125 [9].

While serum proteins have been studied extensively [10], most biomarkers are also expressed by normal tissues and are found at low levels in sera from healthy women. Depending upon antigen expression in ovarian cancers and the rate of release into blood, a substantial volume of ovarian cancer may be required to shed sufficient amounts of protein antigen to elevate serum levels. Smaller volumes of cancer might, however, evoke an immune response resulting in the production of autoantibodies (AAb) against ovarian cancer-associated antigens [11]. Virtually all high grade serous ovarian cancers have mutations of p53 with overexpression of TP53 protein in a majority of cases. Autoantibodies reactive with TP53 are associated with at least 20% of high grade serous ovarian cancers. Titers of anti-TP53 AAb rise eight months prior to CA125 and 22 months prior to clinical diagnosis of cancers that do not elevate serum CA125 levels [4].

The objective of this study is to evaluate 10 candidate antigens [6,12,13,14,15,16,17,18,19,20,21] and 12 potential AAb biomarkers [5,6,22,23,24,25,26,27,28] that have been cited in the world literature for their ability to distinguish patents with early stage ovarian cancer from healthy individuals, seeking biomarkers that would detect cases missed by CA125. Serum specimens have been analyzed from patients with early stage (I–II) and late stage (III–IV) ovarian cancer and from healthy women who participated in the NROSS study. In both discovery and validation sets, a four-biomarker panel of osteopontin, macrophage migration inhibitory factor, IL-8 AAb and CA125 exhibited greater sensitivity than CA125 alone in detecting early stage ovarian cancer.

## 2. Results

### 2.1. Osteopontin (OPN), Macrophage Migration Inhibitory Factor (MIF) and Anti-IL-8 Autoantibodies (AAb) Provide the Highest Sensitivity at 98% Specificity Among 22 Potential Ovarian Cancer Biomarkers

Using Luminex-based Magplex/xMAP assays [4], we measured levels of 10 ovarian cancer associated protein antigens (Osteopontin (OPN), Macrophage Migration Inhibiting Factor (MIF), Human Epididymis Protein 4 (HE4), Leptin, Interleukin-6 (IL-6), Interleukin-8 (IL-8), Tumor Necrosis Factor α (TNFα), Fibroblast Growth Factor 2 (FGF2), Transforming Growth Factor alpha (TGFα), and Vascular Endothelial Growth Factor (VEGF) and 12 autoantibodies reactive with cancer-associated proteins (anti-IL-8, E3 ubiquitin-protein ligase Mdm2 (MDM2), Myc proto-oncogene protein (c-MYC), Tissue-type Plasminogen Activator (PLAT), Epithelial Cell Adhesion Molecule (EpCAM), Homeobox Protein Hox-A7 (HOXA7), Alpha-enolase (ENO1), Protein Disulfide-isomerase (PDI), Endoplasmic Reticulum Chaperone BiP (HSPA5), Heat Shock Cognate 71 kDa Protein (HSPA8), Annexin A2 (ANXA2), and cathepsin D) in sera from 76 patients with early stage (I–II) ovarian cancer, 44 patients with late stage (III–IV) ovarian cancer, and 200 healthy controls who participated in the NROSS study. Characteristics of the cancers are listed in Table 1.

For cost-effective two-stage screening strategies, no more than 2% of patients with elevated serum biomarkers should be referred for imaging with TVS [29,30]. Consequently, we have measured the sensitivity of biomarker candidates at 98% specificity. Among the 22 biomarkers, the highest sensitivity was observed with OPN, MIF and anti-IL-8 AAb (Appendix A). OPN was elevated in 50% of early stage and 46% of late stage cases (Figure 1A). Elevated MIF was detected in 21% of early stage and 39% of late stage ovarian cancers (Figure 1B). Anti-IL-8 autoantibodies were detected in 7% of both early and late stage cases (Figure 1C). CA125, measured with a Roche assay had the highest sensitivity, detecting 65% of early stage and 89% of late stage ovarian cancer cases with a cut-off of 35 U/mL (Figure 1D).

### 2.2. OPN, MIF and Anti-IL-8 Autoantibody Levels Are Elevated in an Independent Validation Set

OPN, MIF, anti-IL-8 AAb and CA125 levels were measured in an independent validation set of sera from 71 patients with early stage ovarian cancer, 45 patients with late stage ovarian cancer and 131 healthy participants in the NROSS trial. Patient characteristics are provided in Table 1. OPN was elevated in 20% of early stage and 16% of late stage disease (Figure 2A). MIF was elevated in 3% of early stage cases and 7% of late stage cases (Figure 2B). Elevated anti-IL-8 AAb levels were detected 10% of sera from early stage and 9% from late stage cases (Figure 2C). CA125 was elevated in 78% of early and 98% of late stage cases (Figure 2D). When control values in the validation and discovery sets were compared, controls in the validation set exhibited higher levels of OPN and MIF, but similar levels of anti-IL-8 AAb, contributing to the decrease in sensitivity observed for OPN and MIF at 98% specificity.

### 2.3. OPN, MIF and IL-8 Autoantibody Complement CA125 for Detecting Ovarian Cancer

Statistical tests based on bootstrap were used to compare AUCs and partial AUCs (pAUCs) to identify the most sensitive combination of biomarkers. Early stage cases had significantly different AUCs between different pairs of models (Table 2).

The model that contained all four biomarkers CA125, OPN, MIF and anti-IL-8 AAb had the highest AUC = 0.985 (95% CI: 0.975–0.995) which differed significantly (*p* < 0.001) from the AUC = 0.885 (95% CI: 0.833–0.937) for CA125 alone. Late stage cases also had different AUC values with different combinations of biomarkers. A combination of CA125, OPN and MIF had the highest AUC = 0.999 (95% CI: 0.998–1.000). When assessing the receiver operating characteristic (ROC) while combining all four markers, our AUC was 0.985 and 0.999 in early and late stage subjects respectively (Figure 3A,B). We then applied two models that had shown the best performances (model 1: CA125, OPN and MIF; model 2: CA125, OPN, MIF and IL-8AAb) to an independent validation data set for early and late stage samples. The validation models had similar AUC and pAUC (Figure 3C,D and Table 3). In early stage disease the four-biomarker combination had a significantly higher (*p* = 0.015) AUC = 0.974 (95% CI 0.957–0.991) than the AUC = 0.947 (95% CI: 0.915–0.978) for CA125 alone. In the discovery set, addition of the three biomarkers improved detection of early stage cancers from 65% with CA125 alone to 82%. In the validation set, the addition of the three biomarkers also improved detection for early stage disease from 78% with CA125 alone to 85%.

## 3. Discussion

In this study we have identified three biomarkers—OPN, MIF and anti-IL-8 AAb—that detect early stage ovarian cancers missed by CA125. Addition of these three biomarkers to CA125 achieved a sensitivity of 82–85% for detecting early stage ovarian cancer compared to 65–78% with CA125 alone in two independent data sets. While OPN and MIF have previously been cited as potential biomarkers for ovarian cancer [31,32], in this study we have had the opportunity for the first time to evaluate OPN and MIF in large numbers of patients with early stage disease.

Complementarity between CA125 and OPN, MIF and anti-IL-8 AAb was demonstrated by ROC analysis of both the discovery set and the validation sets of sera from patients with early stage ovarian cancer. The added values of the three biomarkers were most readily apparent in the discovery set where CA125 detected 65% of the cases and the four-biomarker panel 82%. In the validation set, CA125 alone detected 78% of early stage cases, a much higher fraction than anticipated, decreasing the apparent impact of other biomarkers. Given the greater sensitivity of CA125 for the serous histotype of epithelial ovarian cancer, detection of a larger fraction of early stage cancers with CA125 alone may relate to the larger fraction of serous cancers in the validation set (59%) than in the discovery set (36%). Use of the four biomarkers increased detection in the validation set to 85%, but the greatest contribution was provided by OPN.

OPN is a glycophosphoprotein that is secreted into body fluids by normal osteoblasts, arterial smooth muscle cells, different epithelial cells, activated T cells and macrophages, but can also be overexpressed by cancers that arise at different sites, including ovarian cancer [33,34]. Functionally, OPN can promote ovarian cancer growth in cell culture and in vivo by activating the PI3K/AKT/HIF-1α signaling pathway; OPN expression positively correlates with clinical stage, histological grade and lymph node metastasis [35,36,37]. In earlier studies, we have shown that OPN can be detected in CA125 negative ovarian cancers at the level of tissue expression, consistent with our observations with circulating OPN in this report [32].

MIF was one of the first cytokines to be described as a product of activated T lymphocytes that inhibited the random migration of cultured macrophages [38]. MIF is also a key regulator of immune and inflammatory responses and is produced by a range of cells and tissues [39]. MIF expression can increase during the evolution of several malignances [40]. MIF has been implicated in the angiogenic switch during progression of early cancers and plays a role in macrophage-induced ovarian cancer cell invasiveness [41,42]. Furthermore, Serum levels of MIF have been used as part of algorithms to detect disease in women with BRCA1/2 mutations and for distinguishing malignant from benign disease [43].

IL-8, also known as C-X-C motif ligand (CXCL) 8 (CXCL8), is a small soluble protein that belongs to the CXC chemokine family [44]. Originally identified as a potent neutrophil activator and chemotactic factor, IL-8 is secreted by activated monocytes and macrophages and is dysregulated in a wide variety of solid cancers including ovarian cancer [45]. It is reported to contribute to multiple hallmarks of cancer, including increased proliferation, angiogenesis, invasion, and metastases [46,47,48]. While circulating IL-8 was not found to be a useful biomarker for early stage ovarian cancer, anti-IL-8 AAb detected at least one cancer missed by the other three markers in early and late stage. Why AAb develop against IL-8 is not well understood.

Previously, we reported that a four-biomarker panel comprising CA125, HE4, CA72-4 and MMP-7 showed strong diagnostic performance and significant improvement over the use of CA125 alone in distinguishing patients with early stage ovarian cancer from healthy women [49]. In subsequent studies, MMP-7 failed to detect cases missed by CA125, whereas HE4 and CA72.4 detected 16% of cases missed by CA125 in the UKCTOCS study [9]. The value of HE4 and CA72.4 were confirmed in the European Prospective Investigation into Cancer (EPIC) study [50]. Using a Luminex based assay, however, levels of HE4 were elevated in only 12% of early stage ovarian cancers and 36% of advanced stage disease. Moreover, the same cases could be identified with OPN, MIF and anti-IL-8 AAb. Among the three additional biomarkers evaluated in this report, OPN was the most robust for detecting early stage disease in cases missed by CA125. Future studies will evaluate potential integration of OPN with CA125, HE4 and CA72-4.

## 4. Materials and Methods

### 4.1. Patient Serum Samples

A discovery set of preoperative sera from 76 patients with stage I–II and 44 patients with stage III–IV invasive epithelial ovarian cancer were obtained from the MD Anderson Gynecologic Cancer Tissue Bank. Sera from 200 healthy controls who had participated in the NROSS were also analyzed. An independent validation set included preoperative sera from 71 patients with stage I–II ovarian cancer, 45 preoperative sera from patients with stage III–IV ovarian cancer that had been drawn, processed and frozen more recently than the discovery set and stored in the same bank. Additional sera from 131 healthy controls were obtained from participants in the NROSS. This study was approved by the University of Texas MD Anderson Cancer Center ethical committee on the date 02/11/2019, ethical code number (LAB02-188). All participants had provided consent for use of samples in ethically approved studies.

### 4.2. Immunoassay Components

MagPlex microspheres, xMAP antibody coupling kits and reagents for the immunoassay were purchased from Luminex Corp. (Austin, TX, USA). Recombinant human IL-8 (CXCL8) was purchased from PeproTech (Rocky Hill, NJ, USA). Human MDM2 protein with a His-tag, human PLAT/TPA protein with 6His-tag, human MYC/c-myc protein and human TACSTD1/EPCAM protein with a His-tag were obtained from LifeSpan BioSciences, lnc. (Seattle, WA, USA). MYC/DDK-tagged human HOXA7 protein was obtained from Creative Biomart (Shirley, NY, USA). Human Enolase-1 protein (ENO1), Human Annexin A2 protein (ANXA2), human protein disulfide isomerase (PDI) with his tag, human heat shock 70 kDa protein (HSPA5) with his tag, human heat shock 70 kDa protein-8 (HSPA8) with his tag were purchased from ProSpec-Tany TechnoGene Ltd. (Ness–Ziona, Israel). Human Cathepsin-D protein (CTSD) was obtained from Enzo Life Sciences, lnc. (Farmingdale, NY, USA). Biotin-conjugated mouse anti-IL-8 antibody and R-phycoerythrin conjugated F (ab’) 2-Donkey anti-Rabbit IgG antibody were obtained from Invitrogen (Frederick, MD, USA). Biotin-conjugated mouse anti-MDM2 antibody was purchased from R&D Systems (Minneapolis, MN, USA). Biotin-conjugated rabbit anti-PLAT antibody was purchased from LifeSpan BioSciences, lnc. Monoclonal mouse anti-c-Myc antibody was obtained from Thermo Fisher Scientific (Rockford, IL, USA). Biotin-conjugated mouse anti-CTSD antibody, biotin-conjugated mouse anti-ANXA2 antibody and biotin-conjugated mouse anti-ENO1 antibody were obtained from Novus Biologicals, LLC (Centennial, CO, USA). Biotin-conjugated mouse anti-HSC70 antibody was purchased from Enzo Life Sciences, Inc. Biotin-conjugated mouse anti-His antibody was obtained from Abcam (Cambridge, MA, USA). Rabbit anti-Myc-Tag antibody was purchased from Cell Signal Technology (Boston, MA, USA). Biotin-conjugated goat anti-human IgG Fcγ fragment and anti-mouse IgG antibody were obtained from were obtained from Jackson ImmunoResearch (West Grove, PA, USA). Streptavidin-R-phycoerythrin (SAPE) was purchased from Life Technologies (Carlsbad, CA, USA). R-phycoerythrin (PE) conjugated goat anti-human IgG antibody was purchased from Southern Biotech (Birmingham, AL, USA).

### 4.3. Antigen Immunoassays

Enzyme-linked immunosorbent assays (ELISA) for 10 potential ovarian cancer serum markers (OPN, MIF, Leptin, IL-6, IL-8, TNFa, FGF2, HE4, TGFa, and VEGF) were performed using MILLIPLEX MAP human circulating cancer biomarker magnetic bead panel kits purchased from MilliporeSigma (Burlington, MA, USA). The median fluorescent intensity (MFI) was measured by the Luminex MAGPIX system (Luminex Corp., Austin, TX, USA) with a minimum of 50 beads read per well. The data were acquired and analyzed by xPONENT software version 4.2 (Luminex Corp.)

### 4.4. Autoantibody Immunoassays

Immunoassays for detecting AAbs against 12 antigens (IL-8, MDM2, c-MYC, PLAT, EpCAM, HOXA7, ENO1, PDI, HSPA5, HSPA8, ANXA2, and cathepsin D) were developed using the MagPlex/xMAP technology (Luminex Corp.). In brief, protein antigens were coupled to one million microspheres with an xMAP antibody coupling kit as specified by the manufacturer. Coupling efficiency was confirmed based on the xMAP Cookbook, thirs version (Luminex Corp.). For each assay, a suspension of antigen–microspheres was prepared by diluting the coupled microsphere stocks (one million beads/mL) to a final concentration of 50 beads/μL in phosphate buffered saline (PBS) (Corning. NY. US). Serum samples (2 μL) for assay were diluted in 48 μL PBS buffer. Serum samples were added to 96-well polystyrene microplates and incubated with protein-coupled microspheres for 1 h at room temperature with gentle shaking. Beads were washed twice with PBS with 0.05% Tween 20 (Acros Organics, Geel, Belgium) (PBST) by clipping onto a magnetic plate separator for one minute and the liquid was discarded. Microspheres were then incubated with biotinylated goat anti-human IgG for 30 min at room temperature protecting from light with gentle shaking. Plates were washed twice with a PBST buffer and microspheres were incubated with a fluorescence reporter SAPE for 30 min at room temperature, protecting from light with gentle shaking. After a final wash, the beads were re-suspended in a PBS buffer and fluorescence measured on the MAGPIX system with a minimum 50 beads read per well. The data were acquired and analyzed by the xPONENT software version 4.2.

For anti-IL8 assays, conditions were further optimized. A suspension of IL-8 antigen-microspheres was diluted to a final concentration of 50 beads/μL in assay buffer (AB), which contained 1% bovine serum albumin (BSA) and 0.05% Tween 20 in PBS. Serum samples (2 μL) for assay were diluted in 198 μL AB, from which was transferred 25 μL diluted sample to 96-well polystyrene microplates and incubated with 25 μL IL-8-coupled microspheres for 2 h at room temperature with gentle shaking. Beads were washed twice with AB by clipping onto a magnetic plate separator for one minute and discarding the liquid. Microspheres were then incubated with 25 μL goat anti-human IgG antibody with PE (4 ug/mL) for 30 min at room temperature protecting from light with gentle shaking. After a final wash, the beads were re-suspended in AB and fluorescence measured on the MAGPIX system with a minimum of 50 beads read per well. The data were acquired and analyzed by xPONENT software version 4.2.

### 4.5. CA125 Immunoassay

CA125 levels were assayed with Roche kits on the Elecsys platform (Indianapolis, IN, USA).

### 4.6. Statistical Analysis

Analysis was performed separately for early and late stage data sets. CA125 detected ovarian cancer cases with a cut-off of 35 U/mL. Other biomarkers were dichotomized based on cut-offs that achieved 98% specificity. Logistic regression models were used to assess the discrimination of combined markers and to assess if adding new biomarkers can have additional benefit to discriminate cases and controls. AUC, sensitivity at specificity of 98%, and partial AUC (pAUC) were calculated for each logistic model. Bootstrapping with 1000 bootstrap samples was used to compute 95% confidence intervals for the AUC and pAUC, as well as P values for testing the differences in these values. All statistical analyses performed using R software version 3.5.1 (R Foundation for Statistical Computing, Vienna, Austria).

## 5. Conclusions

Our study demonstrated that OPN, MIF and IL8 AAb can enhance the sensitivity of CA125 to distinguish ovarian cancer patients from health controls at high specificity. The development of reliable panels of tests that can detect cancer at an early and curable stage is worth further investigation.

## Figures and Tables

**Figure 1 cancers-11-00596-f001:**
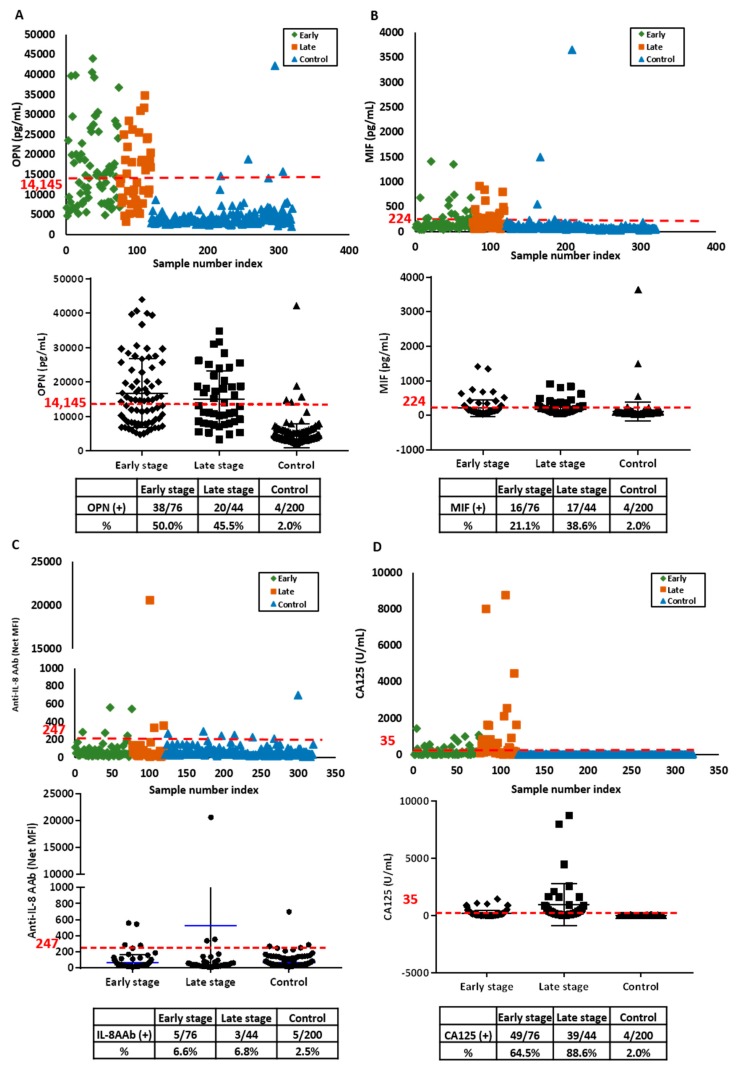
Serum levels of osteopontin (OPN), macrophage inhibitory factor (MIF) anti-IL-8 autoantibody and CA125 are elevated in the discovery set from ovarian cancer patients and healthy controls. (**A**) OPN antigen; (**B**) MIF antigen; (**C**) anti-IL-8 autoantibodies (AAb) and (**D**) CA125 antigen. In the upper panels, each symbol represents the average of duplicate serum samples from individual early stage ovarian cancer cases (green), late stage ovarian cancer cases (red) or controls (blue). The middle panels contain a scatter plot where the blue line represents the median value in early and late stage case and control groups, respectively. The red dashed lines in each plot represent the cut-off value at 98% specificity for OPN, MIF and anti-IL-8 AAb, and cut-off value of 35 U/mL for CA125. The lower panel is a table displaying the sensitivity for early and late stage ovarian cancer cases and controls near 98% specificity.

**Figure 2 cancers-11-00596-f002:**
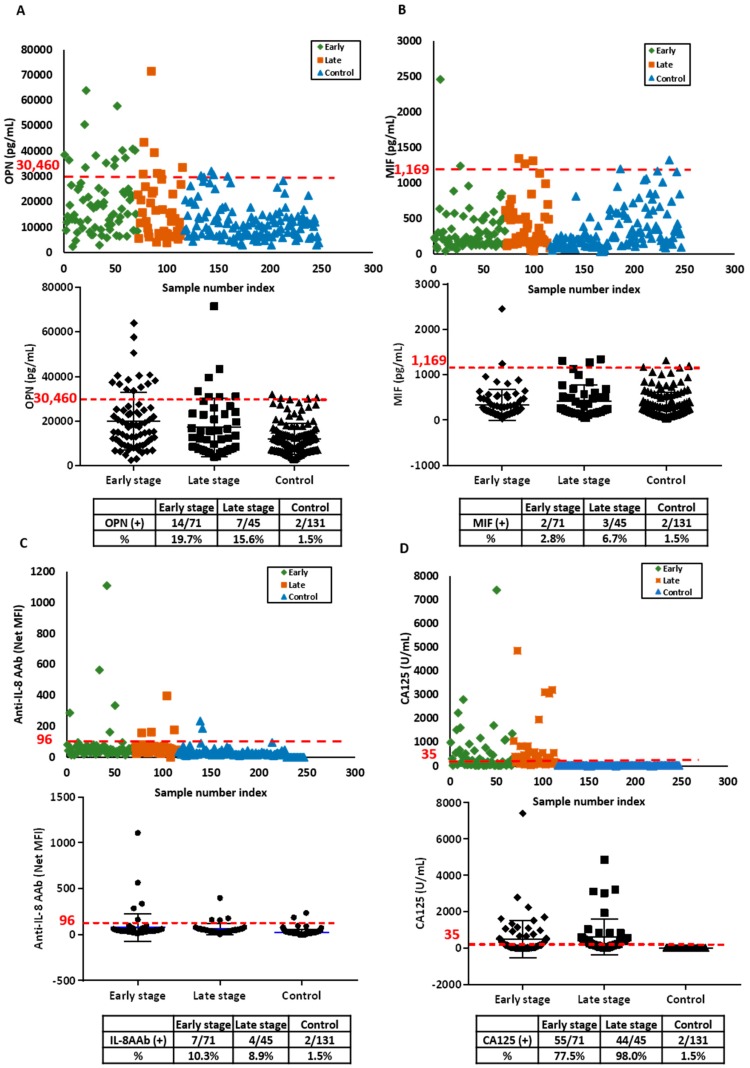
Serum levels of OPN, MIF anti-IL-8 autoantibody and CA125 are elevated in the validation set from ovarian cancer patients and healthy controls. (**A**) OPN antigen; (**B**) MIF antigen; (**C**) anti-IL-8 autoantibody and (**D**) CA125 antigen. In the upper panels, each symbol represents the average of duplicate serum samples from individual early stage ovarian cancer cases (green), late stage ovarian cancer cases (red) or controls (blue). The middle panels contain a scatter plot where the blue line represents the median value in early and late stage case and control groups, respectively. The red dashed lines in each plot represent the cut-off value at 98% specificity for OPN, MIF and anti-IL-8 AAb, and cut-off value of 35 U/mL for CA125. The lower panel is a table displaying the sensitivity for early and late stage ovarian cancer cases and controls near 98% specificity.

**Figure 3 cancers-11-00596-f003:**
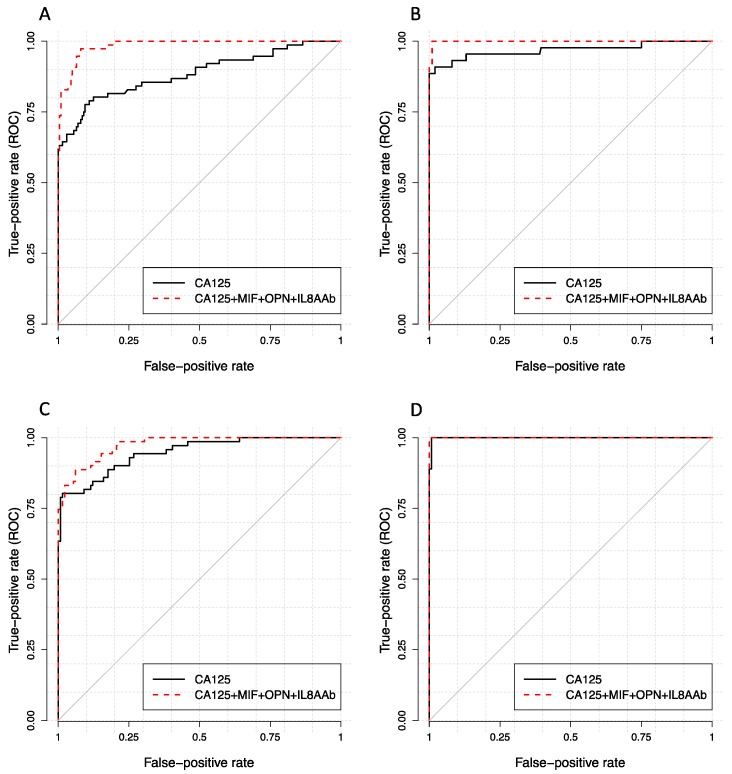
Receiver operating characteristic (ROC) curves for a combination of CA125, OPN, MIF and anti-IL-8 autoantibody biomarkers. (**A**) Early stage cases of discovery set. (**B**) Late stage cases of discovery set. (**C**) Early stage cases of validation set. (**D**) Late stage cases of validation set. A combination of CA125, OPN, MIF and IL-8 AAb levels had a significantly greater AUC (0.985) than did CA125 alone in early stage cases of discovery set (AUC = 0.885, *p* < 0.001). Compare to CA125 alone (AUC = 0.947), the combination also made a significant difference (AUC = 0.974, *p* = 0.015) in early stage cases of validation set.

**Table 1 cancers-11-00596-t001:** Characteristics of the patient population.

Sample set	Stage	No.	Histology					Age (Years)	
			Serous	Endometrioid	Clear Cell	Mucinous	Other	Median	Mean
**Discovery set**	I	50	15	13	7	7	8	56	55
	II	26	12	4	2	2	6	61	60
	III	34	33	0	1	0	0	63	61
	IV	10	9	0	0	0	1	60	58
	Healthy	200						62	63
**Discovery set**	I	41	19	5	4	6	7	55	54
	II	30	23	2	1	0	4	60	61
	III	39	39	0	0	0	0	62	61
	IV	6	6	0	0	0	0	54	59
	Healthy	131						66	66

**Table 2 cancers-11-00596-t002:** Comparison of AUC and pAUC for biomarker models in the discovery set.

Stage	Marker	AUC	pAUC
Early stage	CA125	0.885 (0.833, 0.937)	0.013 (0.011, 0.015)
	CA125 MIF OPN	0.985 * (0.974, 0.996)	0.015 (0.012, 0.018)
	CA125 MIF OPN IL8AAb	0.985 # (0.975, 0.995)	0.015 (0.012, 0.018)
Late stage	CA125	0.969 (0.931, 1.000)	0.018 (0.016, 0.020)
	CA125 MIF OPN	0.999 (0.998, 1.000)	0.019 (0.018, 0.020)
	CA125 MIF OPN IL8AAb	0.999 (0.998, 1.000)	0.019 (0.017, 0.020)

Compared to CA125 alone, * *p* < 0.001; # *p* < 0.001.

**Table 3 cancers-11-00596-t003:** Comparison of AUC and pAUC for biomarker models in the validation set.

Stage	Marker	AUC	pAUC
**Early stage**	CA125	0.947 (0.915, 0.978)	0.015 (0.012, 0.017)
	CA125 MIF OPN	0.955 (0.927, 0.983)	0.015 (0.013, 0.017)
	CA125 MIF OPN IL8AAb	0.974 * (0.957, 0.991)	0.015 (0.013, 0.017)
**Late stage**	CA125	0.999 (0.997, 1.000)	0.999 (0.997, 1.000)
	CA125 MIF OPN	1.000 (1.000, 1.000)	1.000 (1.000, 1.000)
	CA125 MIF OPN IL8AAb	1.000 (1.000, 1.000)	1.000 (1.000, 1.000)

Compared to CA125 alone, * *p* = 0.0153.

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
