# Peer review of "Osteopontin, Macrophage Migration Inhibitory Factor and Anti-Interleukin-8 Autoantibodies Complement CA125 for Detection of Early Stage Ovarian Cancer"

_cancers, 2019, doi:10.3390/cancers11050596_

Round 1

Reviewer 1 Report

This is a very interesting paper which relies on a combinatory evaluation of potential biomarkers for early and late detection of ovarian cancer (OC). In this study, authors identified three reliable markers that could improve the sensibility of CA125 (osteopontin, macrophage migration inhibitory factor and Interleukin-8). Authors conclude that addition of these 3 biomarkers to CA125 achieved sensitivity of 82-85% for detecting early OC stage compared to 65-78% with CA125. Among the three additional biomarkers evaluated in this report, OPN was the most robust for detecting early stage disease in 219 cases missed by CA125.

The paper by Guo et al. should open new avenues into the understanding for developing new strategies for early detection of OC.

The manuscript is overall well-written and methods seem to be rigorously done.

I have only few considerations to this manuscript.  

The study by Simmons et al. (2016) showed that 4-marker panel comprising CA125, HE4, MMP-7, and CA72-4 performed with the highest sensitivity (83.2%) at 98% specificity. So, which is the major difference to the current study in terms of specificity and accuracy during early OC detection? Is there an increment of combining these biomarkers each other?

All analyses were performed in serous OC; however, it would be interesting to see whether the sensibility varied according with tumor subtype (e.g., mucinous, clear cell, endometrioid).   

Were the patients’ features including age, concurrent chemotherapy, tumor recurrence and genetic background taken into account?  

Author Response

Reviewer #1:

Point 1: The study by Simmons et al. (2016) showed that a 4-marker panel comprising CA125, HE4, MMP-7, and CA72-4 performed with the highest sensitivity (83.2%) at 98% specificity. So, which is the major difference to the current study in terms of specificity and accuracy during early OC detection? Is there an increment of combining these biomarkers each other?

Response 1: The specificity in both studies was set at 98% so that only 2% of screened subjects would be referred for transvaginal sonography. In a subsequent study by Simmons et al, Cancer Prevention Research 2019 [Epub ahead of print] that evaluated longitudinal specimens from the UKCTOCS study in the United Kingdom, HE4 and CA72.4 detected 16% of cases missed by CA125, whereas MMP-7 did not detect additional cases. In the current study, CA125, OPN, MIF, and anti-IL-8 AAb were evaluated using a different discovery set of pre-operative serum samples from patients at MD Anderson Cancer Center. In future studies, we plan to test CA125, HE4, CA72.4, osteopontin, M-CSF and anti-IL8 autoantibody on the same set of early stage serum samples to determine whether the combination of biomarkers will detect additional cases.

Point 2: All analyses were performed in serous OC; however, it would be interesting to see whether the sensibility varied according with tumor subtype (e.g., mucinous, clear cell, endometrioid).

Response 2: In fact, early and late stage samples included not only high grade serous ovarian cancers, but also cancers of endometrioid, clear cell and mucinous histotypes as indicated in Table 1.

Discovery set

Early stage

Late stage

Histology

OPN (+)

Number (%)

MIF (+)

Number (%)

IL-8 AAb (+)

Number (%)

Histology

OPN (+)

Number (%)

MIF (+)

Number (%)

IL-8 AAb (+)

Number (%)

Type

Number

Type

Number

Serous

27

16 (42%)

9 (56%)

2 (40%)

Serous

42

18 (90%)

17 (100%)

2 (67%)

Endometrioid

17

7 (18%)

3 (19%)

1 (20%)

Endometrioid

0

0

0

0

Clear cell

9

5 (13%)

1 (6%)

1 (20%)

Clear cell

1

1 (5%)

0

1 (33%)

Mucinous

9

3 (8%)

2 (13%)

0

Mucinous

0

0

0

0

Others

14

7 (18%)

1 (6%)

1 (20%)

Others

1

1 (5%)

0

0

Total

76

100%

100%

100%

Total

44

100%

100%

100%

Validation set

Early stage

Late stage

Histology

OPN (+)

Number (%)

MIF (+)

Number (%)

IL-8 AAb (+)

Number (%)

Histology

OPN (+)

Number (%)

MIF (+)

Number (%)

IL-8 AAb (+)

Number (%)

Type

Number

Type

Number

Serous

42

7 (50%)

1 (50%)

4 (57%)

Serous

42

7 (100%)

3 (100%)

4 (100%)

Endometrioid

7

1 (7%)

0

1 (14%)

Endometrioid

0

0

0

0

Clear cell

6

2 (14%)

0

1 (14%)

Clear cell

1

0

0

0

Mucinous

6

2 (14%)

0

0

Mucinous

0

0

0

0

Others

11

2 (14%)

1 (50%)

1 (14%)

Others

1

0

0

0

Total

71

100%

100%

100%

Total

44

100%

100%

100%

Point 3: Were the patients’ features including age, concurrent chemotherapy, tumor recurrence and genetic background taken into account?  

Response 3: All samples assayed in the current study were obtained pre-treatment at the time of initial diagnosis so concurrent chemotherapy and tumor recurrence were not an issue. Genetic background was not detailed, but no more than 15% of patients with high grade serous cancers should have germ line BRCA1/2 mutations. The ages of patients and controls is indicated in the following table.  Median ages of controls and patients in the Discovery Set were comparable, although the median age of controls in the Validation Set was slightly older.

Stage

Age   (years)

Median

Mean

Discovery   Set

I

56

55

II

61

60

III

63

61

IV

60

58

Healthy

62

63

Validation   Set

I

55

54

II

60

61

III

62

61

IV

54

59

Healthy

66

66

Reviewer 2 Report

The manuscript “Osteopontin, Macrophage Migration Inhibitory 2 Factor and Anti-Interleukin-8 Autoantibodies 3 Complement CA125 for Detection of Early Stage 4 Ovarian Cancer” by Guo et al. describes a panel of biomarkers for the detection of ovarian cancer. The paper identifies the combination of four markers plus one autoantibody as an improvement over the conventional blood marker CA125.

Although the study is somewhat limited in scope (2 data Figures, 2 Tables and 1 Figure with analytics), it points to important possibilities. The two-step approach, moving from a discovery set to a validation set, is a strength. The main weakness is the poor justification of the initial choice of markers (references are cited, but they do not seem to be derived from any kind of a systematic review of the literature) and, more importantly, the selection of the markers to include in the panel. The Supplementary Table would benefit from additional columns with raw data and clear cut-off rules.

Table1 is poorly formatted and needs corrections.

Author Response

Point 1: The main weakness is the poor justification of the initial choice of markers (references are cited, but they do not seem to be derived from any kind of a systematic review of the literature) and, more importantly, the selection of the markers to include in the panel. The Supplementary Table would benefit from additional columns with raw data and clear cut-off rules.

Response 1:

1.    The 6 autoantibodies (anti-IL-8, MDM2, PLAT, EPCAM, HOXA7, c-myc) were chosen that exhibited the highest sensitivity in a recent comprehensive review of the world literature: “Systematic review: Tumor-associated antigen autoantibodies and ovarian cancer early detection” (Gynecologic Oncology 2017; 147:465–480).

2.    The other 6 autoantibodies (ENO1, PDI, HSPA5, HSPA8, ANXA2, and cathepsin D) were chosen from an article regarding “Identification of immunoreactive tumour antigens using free and exosome-associated humoral responses” (Journal of Circulating Biomarkers 2013; https://doi.org/10.5772/57524) that measured patient-derived antibodies which recognized tumor antigens on circulating exosomes.

3.    OPN was chosen from our previous study “Potential markers that complement expression of CA125 in epithelial ovarian cancer” (Gynecologic Oncology 2005; 99:267-277) that identified biomarkers which complement CA125 at the tissue level in ovarian cancers.

4.    The other protein biomarkers (MIF, Leptin, IL-6, IL-8, TNFa, FGF2, HE4, TGFa, and VEGF) were chosen based on the availability of commercially available Millipore kits.

Point 2: The Supplementary Table would benefit from additional columns with raw data and clear cut-off rules. The S1 Table has been revised. Supplementary figures have been provided S1-S6 with the primary data.

Response 2: Data have been provided for the other protein biomarkers at 98% cut-offs in supplementary table 1.

Point 3: Table1 is poorly formatted and needs corrections.

Response 3: We have reformatted Table 1.